# Progressive Mixup Augmented Teacher-Student Learning for Unsupervised Domain Adaptation

## Abstract

Unsupervised Domain Adaptation (UDA) aims to transfer knowledge learned from a labeled source domain to an unlabeled target domain, mostly through learning a domain invariant feature representation. Currently, the best performing UDA methods use category level domain alignment to capture fine-grained information, resulting in significantly improved performance over global alignment. While successful, category level UDA methods suffer from the unreliable pseudo-labels for target data. Additionally, most UDA methods directly adapt from source to target domain without regard for the large domain discrepancy. In this paper, we propose an UDA approach with teacher-student learning where the teacher network is used to provide more reliable target pseudo-labels for the student during training. Furthermore, we use a progressive mixup augmentation strategy which generates intermediate samples that become increasingly target-dominant as training progresses. Aligning the source and intermediate domains allows the model to gradually transfer fine-grained domain knowledge from the source to the target domain while minimizing the negative impact of noisy target pseudo-labels. This progressive mixup augmented teacher-student (PMATS) training strategy along with class subset sampling and clustering based pseudo-label refinement achieves state-of-the-art performance on two public UDA benchmark datasets: Office-31 and Office-Home.

## 1 Introduction

Unsupervised Domain Adaptation (UDA) has become a popular research topic due to its necessity in applying deep learning models to real world scenarios. Often, there exists a domain gap (Quinonero-Candela et al., 2008) between training data and real world testing data that negatively affects model performance during test time. Collecting and labeling data from various domains is impractical due to being both time consuming and labor intensive. Though semi-supervised learning (Berthelot et al., 2019; Sohn et al., 2020) and unsupervised learning (Gidaris et al., 2018) have been studied to improve generalizability to unseen and unlabeled data, it is under the assumption that both the labeled and unlabeled data existed in a similar domain.

UDA specifically tackles the problem of data distribution shift, or domain shift, between the labeled source data and the unlabeled target data through transferring knowledge learned from the source domain to the target domain. UDA methods can mostly be split into two different categories. Adversarial adaptation methods (Ganin & Lempitsky, 2015; Tzeng et al., 2017), inspired by generative adversarial networks (GANs), introduce a domain discriminator to encourage domain confusion in the feature generator through domain-adversarial objectives. This minimizes the gap between source and target distributions through learning domain invariant features. Statistical adaptation methods (Long et al., 2015; 2017) align source and target domain distributions through minimizing a statistical discrepancy measure, such as maximum mean discrepancy (MMD) and joint MMD (JMMD), between the two domains. Most domain adaptation methods in these two categories directly align source and target domain distributions (Ganin & Lempitsky, 2015; Tzeng et al., 2017; Long et al., 2015; 2017) without consideration for the large domain discrepancy, e.g. from synthetic images to real images (Peng et al., 2017) or from art to real images (Venkateswara et al., 2017). Even when Na et al. (2021) and Hua & Guo (2020) attempt to address the large domain gap, they only create

a small number of augmented intermediate domains (4 or less) based on an arbitrarily fixed mixup ratio between source and target images.

Recent domain adaptation research (Zhu et al., 2020; Long et al., 2018; Pei et al., 2018; Kumar et al., 2018) has found performing domain alignment at the category level, taking into account the source and target sample class information, to be more effective than naively learning a global domain shift. Since target data are unlabeled in the UDA setting, category level alignment methods rely on producing pseudo-labels for target samples. These generated pseudo-labels are noisy and unreliable, presenting a problem for deep convolutional neural networks (CNNs) which lack robustness to such pseudo-labels (Morerio et al., 2020; Jiang et al., 2020). Replacing the CNN backbone with a vision transformer (Xu et al., 2021) has been shown to improve model robustness to noisy pseudo-labels.

In this paper, we address the two issues of large domain gap and noisy pseudo-labels mentioned above by proposing a progressive mixup training strategy with teacher-student learning. We construct an intermediate augmented domain that becomes progressively more target-like as training continues. During different periods of training, the intermediate domain has different characteristics. For example, the intermediate domain is more source-like initially, with more reliable label information but lower correlation with target domain. As training progresses, the intermediate domain becomes more target-like, with less reliable label information but higher similarity to target domain. By gradually changing the intermediate domain from source-like to target-like, we're able to train one model that retains the benefits of both perspectives without using an ensemble of models as in Na et al. (2021). For category level subdomain alignment, we use a teacher model to generate pseudo-labels for training our student model. Our teacher model, as a temporal ensemble (Tarvainen & Valpola, 2017) of the student model, produces less noisy target predictions by averaging together the predictions of many student models at previous time steps. Furthermore, we use a clustering based pseudo-label refinement method to obtain more reliable soft pseudo-labels.

We evaluate the performance of our Progressive Mixup Augmented Teacher-Student (PMATS) algorithm on two different UDA benchmarks with varying degrees of domain shift. Experiments prove the effectiveness of our approach since we achieve state-of-the-art performance on both datasets. The contributions of this paper are summarized as follows.

- We propose an algorithm that effectively combines teacher-student learning with a progressive mixup to efficiently bridge the source and target domains utilizing a gradually shifting intermediate domain. This not only effectively combines knowledge from both source-like and target-like perspectives, but also increases the model robustness to noisy pseudo-labels.
- Target predictions from our classifier are further refined through spherical K-means clustering and Gaussian Mixture Model (GMM) to produce more reliable soft pseudo-labels.
- We validate the effectiveness of our approach through extensive ablation studies and evaluation on three standard benchmarks.

## 2 RELATED WORKS

### 2.1 UNSUPERVISED DOMAIN ADAPTATION

Most recent UDA methods are based on domain alignment and discriminative domain-invariant feature learning methods. Statistical methods train a deep adaptation network (DAN) (Long et al., 2015) to minimize the maximum mean discrepancy (MMD) between source and target features over domain specific layers. Building on top of MMD, joint adaptation networks (JAN) (Long et al., 2017) aligns the joint distributions of multiple domain-specific layers using a joint maximum mean discrepancy (JMMD). Weighted MMD (Yan et al., 2017) alleviates class weight bias by assigning class specific weights to source data. Moving from global domain alignment to category level, or subdomain, alignment, deep subdomain adaptation network (DSAN) (Zhu et al., 2020) uses a local maximum discrepancy (LMMD) to align the subdomain distributions. The Contrastive Adaptation Network (CAN) (Kang et al., 2019) uses a similar discrepancy measure called the Contrastive Domain Discrepancy (CDD) which not only minimizes the intra-class subdomain discrepancy, but also maximizes the inter-class subdomain discrepancy in a contrastive manner. Completely replacing the MMD based loss, cross-domain contrastive learning (CDCL) (Wang et al., 2022) instead uses contrastive learning to align source and target distributions through a modified noise-contrastive es-

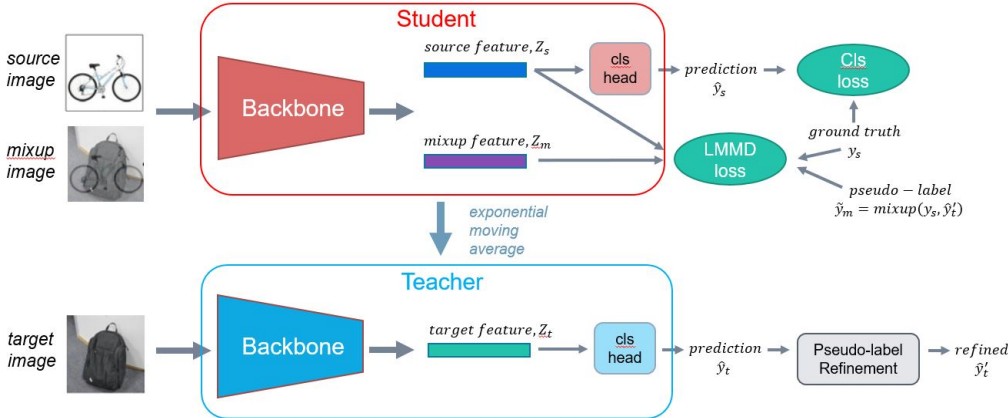

Figure 1: An overview of our proposed pipeline. Source data and augmented mixup data will be input into the student feature extraction backbone while target data will be input into the teacher feature extraction backbone to obtain corresponding embedding features $Z_s$, $Z_m$, and $Z_t$. Target feature $Z_t$ will go through the teacher classification head to obtain target prediction $\hat{y}_t$, which will then go through clustering-based pseudo-label refinement to obtain a refined probability distribution $\hat{y}_t'$. The mixup label $\widetilde{y}_m$, calculated from target refined pseudo-label $\hat{y}_t'$ and source ground truth label $y_s$, will be used in the LMMD loss to align source and mixup feature distributions at the subdomain level. Finally, source feature $Z_s$ will also go through the student classification head to obtain source prediction $\hat{y}_s$ for classification (cross-entropy) loss. The student network weights are updated based on these two loss functions while the teacher network is an exponential moving average of the student network.

timation (infoNCE) loss (Oord et al., 2018), which has the anchor sample from one domain and the corresponding positive and negative samples be from another domain. Adversarial methods, such as the domain adversarial neural network (DANN) (Ganin & Lempitsky, 2015), learn a domain invariant feature representation through back-propagating the reverse gradients of a domain classifier. Building on top of DANN, dual mixup regularized learning (DMRL) (Wu et al., 2020) guides the classifier to enhance consistent predictions between samples and enriches the intrinsic structures of the latent space through two mixup regularizations. FixBi (Na et al., 2021) uses mixup to bridge the large domain gap between the source and target domains by creating two intermediate representations through fixed ratio-based mixup between source and target samples.

## 2.2 PSEUDO-LABELING

For category level domain alignment, generating pseudo-labels (Lee et al., 2013) is a necessity due to the lack of target labels in the UDA setting. For domain adaptation, pseudo-labels have been generated directly from the model prediction (Zhu et al., 2020; Long et al., 2018) for conducting conditional or subdomain distribution alignment. Caron et al. (2018) propose a deep self-supervised method by generating pseudo-labels via K-means cluster, which is also used in Kang et al. (2019). Cross-domain transformer (CDTrans) (Xu et al., 2021) combines both methods by first initializing cluster centroids with weighted K-means clustering using the model probability output. Our pseudo-label refinement method takes this a step further by using the K-means clustering results as initialization for a Gaussian Mixture Model (GMM) to produce a refined soft pseudo-label.

## 2.3 SELF-ENSEMBLING AND MEAN TEACHER

Self-ensembling was originally proposed for generating pseudo-labels in the semi-supervised learning scenario. The temporal ensemble (Laine & Aila, 2016) averaged the predictions of a model over time for a better approximation of the true label compared to the model prediction at a single time. Self-ensembling was extended to the Mean Teacher model (Tarvainen & Valpola, 2017) which directly averaged the model weights rather than the model predictions. A student model would

be trained with cross-entropy loss on labeled data and to match the teacher model predictions on unlabeled data. The Mean Teacher model has been directly used for domain adaptation on image classification (French et al., 2017) and medical imaging segmentation (Perone et al., 2019) tasks, training the student model to match the teacher model prediction for unlabeled target data. Rather than directly performing domain adaption with the Mean Teacher method, we use the better teacher pseudo-labels for subdomain feature alignment.

## 2.4 Vision Transformer

The transformer was proposed in Vaswani et al. (2017) to model sequential data in the field of Natural Language Processing (NLP). It has since been adapted to computer-vision tasks with great success, resulting in transformer based models becoming more and more popular. For example, the Vision Transformer (ViT) (Dosovitskiy et al., 2020) uses sequences of image patches as the transformer input. The Data-efficient Image Transformer (DeiT) (Touvron et al., 2021) introduces a distillation strategy for transformers to help with ViT training. ViT variants have been shown to outperform their CNN counterparts in various tasks beyond image classification, such as object detection (Liu et al., 2021), semantic segmentation (Yuan et al., 2021), and object ReID (He et al., 2021). This trend is also seen in domain adaption for image classification, where the transformer based method (Xu et al., 2021) outperforms other CNN based UDA methods on most benchmarks. We conduct experiments using both CNN and transformer backbones of similar size to show that our method is effective regardless of the backbone architecture.

## 3 Method

In unsupervised domain adaptation, we are given a source domain $D_s = \{(x_i^s, y_i^s)\}_{i=1}^{n_s}$ of $n_s$ source samples and a target domain $D_t = \{(x_j^t)\}_{j=1}^{n_t}$ of $n_t$ target samples where $y_i^s \in R^C$ is the one-hot label of $x_i^s$ and $C$ is the number of classes. $D_s$ and $D_t$ are sampled from distributions $p$ and $q$ respectively, where $p \neq q$. Our goal in the UDA scenario is to train a network on labeled source domain data $x^s$ and unlabeled target domain data $x^t$ to make accurate predictions $\hat{y}^t$ on $x^t$.

In this section, we will go over the components of our PMATS algorithm as shown in Figure 1.

### 3.1 Teacher Network

The teacher network is identical to the student network in architecture and is updated at iteration $t$ based on the equation

$$\theta_t^{'} = \theta_{t-1}^{'}\alpha + \theta_t(1 - \alpha), \tag{1}$$

where $\theta^{'}$ and $\theta$ are the teacher and student model parameters respectively and $\alpha$ is the exponential moving average (EMA) decay.

### 3.2 Clustering-based Pseudo-label Refinement

Given target features $z_j^t$ from our feature generator backbone and target probability distribution $\hat{y}_j^t$ from our classifier as the initial pseudo-label, we perform a series of clustering operations to refine the pseudo-label as shown in Figure 2. This is based on the assumption that even with domain shift, target features of the same class tend to be close together. We use this assumption to refine our pseudo-labels based on the K-means clustering of target samples. K-means clustering gives us a hard label that may not be suitable for samples close to multiple cluster centers. Therefore, we use a GMM to convert the hard label to a soft probability distribution. Details of our pseudo-label refinement process are discussed in Appendix B.

### 3.3 Subdomain Alignment

To perform the category level domain alignment, we use the local maximum mean discrepancy (LMMD) loss proposed in Zhu et al. (2020), which takes advantage of the target probability distribution for calculating discrepancy, unlike the similiar CDD (Kang et al., 2019) loss, which uses hard target pseudo-labels. We will go over the concept of MMD before formulating the LMMD loss as an extension of MMD.

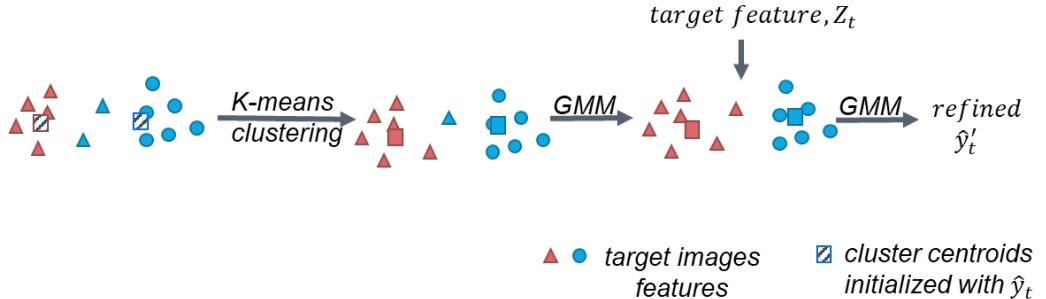

Figure 2: Pseudo-label refinement. Target predictions from classifier head will be used to initialize cluster centroids for K-means clustering. The results of the K-means clustering will then be used to initialize a GMM for optimization with the EM algorithm. Finally, the GMM will output a refined pseudo-label $\hat{y}_t'$ given target feature $Z_t$

### 3.3.1 MAXIMUM MEAN DISCREPANCY

Maximum mean discrepancy (MMD) (Gretton et al., 2012) is a kernel two-sample test, which evaluates the null hypothesis that $p = q$ based on the idea that if two distributions are identical, then all their statistics should be the same. MMD defines the difference between two distributions as the difference in their means in the reproducing kernel Hilbert space (RKHS) $H$:

$$d_H(p,q)\|E_p[\phi(x^s)] - E_q[\phi(x^t)]\|_H^2 \tag{2}$$

where $\phi(*)$ is some functional mapping that maps the original samples to the RKHS with some kernel. In practice, MMD is estimated with the empirical kernel mean embeddings as:

$$\hat{d}_H(p,q) = \left\|\frac{1}{n_s}\sum_{x_i \in D_s}\phi(x^s) - \frac{1}{n_t}\sum_{x_j \in D_t}\phi(x^t)\right\|_H^2. \tag{3}$$

### 3.3.2 LOCAL MAXIMUM MEAN DISCREPANCY

To align the distributions of relevant subdomains rather than the global source and target distributions, LMMD (Zhu et al., 2020) calculates a difference values for each of the $C$ categories using a class weight $w^c$, representing the probability that the sample belongs to class c.

$$\hat{d}_H(p,q) = \frac{1}{C}\sum_{c=1}^{C}\left\|\sum_{x_i \in D_s}w_i^{sc}\phi(x^s) - \sum_{x_j \in D_t}w_j^{tc}\phi(x^t)\right\|_H^2, \tag{4}$$

Where weight $w_i^c$ for sample $x_i$ is calculated as:

$$w_i^c = \|\frac{y_{ic}}{\sum_{(x_j,y_j)\in D}y_{jc}}, \tag{5}$$

where $y_{ic}$ is the c-th entry for vector $y_i$. For source samples, one-hot true label $y_i^s$ is used to calculate weight $w_i^{sc}$. For target samples, we use the soft pseudo-label $\hat{y}_j^t$ to calculate weight $w_j^{tc}$.

Note that in our PMATS algorithm, we align the source distribution $p$ with an augmented intermediate distribution $q'$ rather than with the target distribution $q$ directly. To adapt the feature layers, we use the features $z_i^s$ and $z_j^m$ corresponding to source sample $x_i^s$ and mixup sample $x_j^m$ respectively. The mixup sample $x_j^m$ and corresponding mixup label $y_j^m$ are obtained through the progressive mixup augmentation covered in the following section. Finally, thanks to the characteristics of the

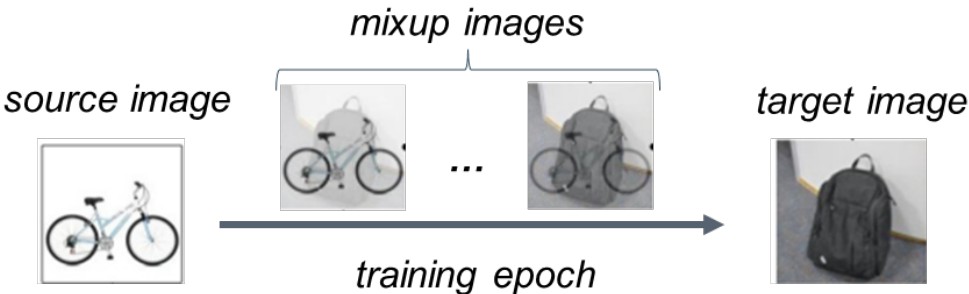

Figure 3: Progressive Mixup Augmentation. Augmented mixup images will gradually transition from source-like to target-like as training progresses.

RKHS, we can reformulate Eq. (4) as

$$
\hat{d}_H(p, q') = \frac{1}{C} \sum_{c=1}^{C} \left[ \sum_{i=1}^{n_s} \sum_{j=1}^{n_s} w_i^{sc} w_j^{sc} k(\phi(z_i^s), \phi(z_j^s)) \right.
$$
$$
+ \sum_{i=1}^{n_m} \sum_{j=1}^{n_m} w_i^{mc} w_j^{mc} k(\phi(z_i^m), \phi(z_j^m)) \tag{6}
$$
$$
\left. -2 \sum_{i=1}^{n_s} \sum_{j=1}^{n_m} w_i^{sc} w_j^{mc} k(\phi(z_i^s), \phi(z_j^m)) \right].
$$

During training, we combine LMMD loss $L_{lmmd}$ with cross-entropy loss $L_{CE}$ on the labeled source data.

### 3.4 PROGRESSIVE MIXUP AUGMENTATION

Mixup (Zhang et al., 2017) is a data augmentation method generally used to increase neural network robustness to noisy pseudo-labels. In the context of UDA, mixup has been used to construct virtual samples by taking the convex combination of labeled and unlabeled data (Na et al., 2021; Wu et al., 2020). Most methods use a mixup ratio $\lambda$ sampled from the beta distribution: $\lambda \sim Beta(\alpha, \alpha)$, where $\alpha$ is a hyperparameter. FixBi (Na et al., 2021) instead uses two fixed ratios to avoid randomness. PrDA Hua & Guo (2020) uses four fixed ratios for what they call progressive domain augmentation. We also sample our mixup ratio from the beta distribution, but with different hyperparameters:

$$
\lambda \sim Beta(d + e\alpha, \beta), \tag{7}
$$

where $d, \alpha$, and $\beta$ are hyperparameters and $e$ is the epoch number. By keeping $\beta$ and $d$ fixed, we use $\alpha$ to decide how quickly we transition from source-like to target-like and $d$ controls how close the mixup sample is to the source sample initially. Given a pair of samples $(x^s, x^t)$ and their labels $(y^s, \hat{y}'^s)$ from the source and target domains respectively, the corresponding mixup sample and label is given as

$$
\widetilde{x}^m = (1 - \lambda)x^s + \lambda x^t, \tag{8}
$$

$$
\widetilde{y}^m = (1 - \lambda)y^s + \lambda \hat{y}'^t. \tag{9}
$$

Another key difference between our method and PrDA is that PrDa only creates virtual samples $x^m$ and not their corresponding label $y^m$ as they perform domain adaptation through multiple subspace alignment on the Grassmann manifold. Our progressive mixup augmentation is visualized in Figure 3

---

**Algorithm 1:** PMATS Training Procedure

---

**Input** : unlabeled target dataset $D_t = X_t$, source dataset $D_s = (X_s, Y_s)$, student model $f$, teacher model $f'$, max epochs $E$, iterations per epoch $K$, warmup epochs $W$, set of all classes $C$

**for** $e = 1$ *to* $E$ **do**

    Initialize cluster centers with target class predictions using Eq. (10(;

    Perform K-means clustering on target data $X_t$, obtain intermediate pseudo-labels $y^{*t}$;

    Fit GMM model to target data $X_t$ while using $y^{*t}$ to initialize parameters based on Eqs. (14), (15), (16), (17) to obtain refined pseudo-labels $y'^t$;

    **for** $k = 1$ *to* $K$ **do**

        Randomly choose a subset of classes $C'$ from the set of all classes $C$;

        **if** $e \leq W$ **then**

            | *Sample batches $(x_i^s, y_i^s)$ and $(x_j^t, y_j'^t)$ belonging to classes in $C$ from $D_s$ and $D_t$;*

        **end**

        **else**

            | *Sample batches $(x_i^s, y_i^s)$ and $(x_j^t, y_j'^t)$ belonging to classes in $C'$ from $D_s$ and $D_t$;*

        **end**

        *Calculate loss = $L_{CE} + \lambda_t L_{lmmd}$ based on Eq. (6);*

        *Back-propagate and update model $f$;*

        *Update model $f'$ based on Eq. (1);*

    **end**

**end**

---

### 3.5 IMPLEMENTATION DETAILS

The training process of our PMATS algorithm is show in Algorithm 1. For further implementation settings, please see Appendix A.

## 4 EXPERIMENTS

We evaluate our proposed method PMATS on two domain adaptation benchmarks and compare with sate-of-the-art UDA methods. Additionally, we validate the contribution of each of our components through extensive ablation studies.

### 4.1 DATASETS

**Office-31** (Saenko et al., 2010) is one of the most popular datasets for real-world domain adaptation, containing a total of 4,110 images of 31 different categories. Office-31 had 3 different domains: the Amazon (A) domain with 2817 images, the Webcam (W) domain with 795 images, and the DSLR (D) domain with 498 images. Despite being popular, the relatively small size of the dataset and similar domains makes it less useful for comparing SoTA methods when most can achieve similarly high accuracies.

**Office-Home** (Venkateswara et al., 2017) is a newer dataset consisting of 15,588 images over 65 different categories. Office-Home has 4 different domains: the Artistic (A) domain, the Clipart (C) domain, the Product (P) domain, and the Real-World (R) domain. Besides being much larger than Office-31, the domain combinations in Office-Home offer a nice mix of easy and difficult adaptation tasks.

### 4.2 FEATURE VISUALIZATION

We use t-SNE (Van der Maaten & Hinton, 2008) to visualize features from the source and target domain before and after alignment in Figure 4. Before alignment, we can see that the source and target features are separated, indicating a domain gap. After alignment with PMATS, the target features mostly form compact clusters around the source features, except for the target feature near the center which still remains spread out. These spread out feature are difficult to classify and thus

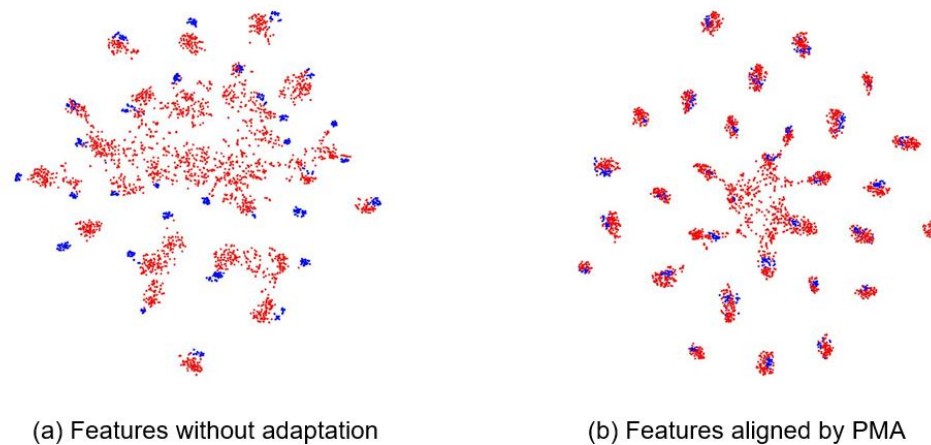

(a) Features without adaptation          (b) Features aligned by PMA

Figure 4: The t-SNE (Van der Maaten & Hinton, 2008) visualization of features before and after alignment on task $A \rightarrow W$. Red points are source samples and blue points are target samples

is not forced by our LMMD loss into a distinct cluster. By using soft pseudo-labels from a GMM rather than a hard pseudo-label from K-means clustering, we are able to avoid aligning these hard to classify samples with the incorrect subdomain distribution.

Table 1: Comparison with SoTA on Office-31 dataset. The best performance is marked in **bold**. * indicates that Deit-small was used as backbone instead of ResNet50.

| Method | $A \rightarrow D$ | $A \rightarrow W$ | $D \rightarrow A$ | $D \rightarrow W$ | $W \rightarrow A$ | $W \rightarrow D$ | Avg |
|---|---|---|---|---|---|---|---|
| ResNet50 (He et al., 2016) | 68.9 | 68.4 | 62.5 | 96.7 | 60.7 | 99.3 | 76.1 |
| DANN (Ganin & Lempitsky, 2015) | 79.7 | 82.0 | 68.2 | 96.6 | 67.4 | 99.1 | 82.2 |
| CDAN+E (Long et al., 2018) | 92.9 | 94.1 | 71.0 | 98.6 | 69.3 | **100.0** | 87.7 |
| DSAN (Zhu et al., 2020) | 90.2 | 93.6 | 73.5 | 98.3 | 74.8 | **100.0** | 88.4 |
| CAN (Kang et al., 2019) | 95.8 | 94.5 | 78.0 | 99.1 | 77.0 | 99.8 | 90.6 |
| SRDC (Tang et al., 2020) | 95.8 | 95.7 | 76.7 | 99.2 | 77.1 | **100.0** | 90.8 |
| RSDA-MSTN (Gu et al., 2020) | 95.8 | 96.0 | 77.4 | **99.3** | 78.9 | **100.0** | 91.1 |
| FixBi (Na et al., 2021) | 95.0 | 96.1 | 78.7 | **99.3** | 79.4 | **100.0** | 91.4 |
| Deit-S* (Touvron et al., 2021) | 87.6 | 86.9 | 74.9 | 97.7 | 73.5 | 99.6 | 86.7 |
| CDTrans-S* (Xu et al., 2021) | 94.6 | 93.5 | 78.4 | 98.2 | 78.0 | 99.6 | 90.4 |
| PMATS | 95.6 | 96.4 | 78.7 | 99.0 | 79.2 | **100.0** | 91.5 |
| PMATS* | **97.4** | **97.7** | **80.4** | 99.0 | **80.7** | 99.8 | **92.5** |

Table 2: Comparison with SoTA on Office-Home dataset. The best performance is marked in **bold**. * indicates that Deit-small was used as backbone instead of ResNet50.

| Method | $A \rightarrow C$ | $A \rightarrow P$ | $A \rightarrow W$ | $C \rightarrow A$ | $C \rightarrow P$ | $C \rightarrow R$ | $P \rightarrow A$ | $P \rightarrow C$ | $P \rightarrow R$ | $R \rightarrow A$ | $R \rightarrow C$ | $R \rightarrow P$ | Avg |
|---|---|---|---|---|---|---|---|---|---|---|---|---|---|
| ResNet50 (He et al., 2016) | 44.7 | 66.3 | 74.3 | 51.8 | 61.9 | 63.6 | 52.4 | 39.1 | 71.2 | 63.8 | 45.9 | 77.2 | 59.4 |
| DANN (Ganin & Lempitsky, 2015) | 45.6 | 59.3 | 70.1 | 47.0 | 58.5 | 60.9 | 46.1 | 43.7 | 68.5 | 63.2 | 51.8 | 76.8 | 57.6 |
| CDAN+E (Long et al., 2018) | 54.6 | 74.1 | 78.1 | 63.0 | 72.2 | 74.1 | 61.6 | 52.3 | 79.1 | 72.3 | 57.3 | 82.8 | 68.5 |
| DSAN (Zhu et al., 2020) | 54.4 | 70.8 | 75.4 | 60.4 | 67.8 | 68.0 | 62.6 | 55.9 | 78.5 | 73.8 | 60.3 | 83.1 | 67.6 |
| SRDC (Tang et al., 2020) | 52.3 | 76.3 | 81 | 69.5 | 76.2 | 78 | 68.7 | 53.8 | 81.7 | 76.3 | 57.1 | 85 | 71.3 |
| RSDA-MSTN (Gu et al., 2020) | 53.2 | 77.7 | 81.3 | 66.4 | 74 | 76.5 | 67.9 | 53 | 82 | 75.8 | 57.8 | 85.4 | 70.9 |
| FixBi (Na et al., 2021) | 58.1 | 77.3 | 80.4 | 67.7 | 79.5 | 78.1 | 65.8 | 57.9 | 81.7 | 76.4 | 62.9 | 86.7 | 72.7 |
| Deit-S* (Touvron et al., 2021) | 55.6 | 73.0 | 79.4 | 70.6 | 72.9 | 76.3 | 67.5 | 51.0 | 81.0 | 74.5 | 53.2 | 82.7 | 69.8 |
| CDTrans-S* (Xu et al., 2021) | 60.6 | 79.5 | 82.4 | **75.6** | 81.0 | **82.3** | 72.5 | 56.7 | 84.4 | 77.0 | 59.1 | 85.5 | 74.7 |
| PMATS | 57.8 | 78.1 | 81.6 | 66.5 | 78.9 | 80.4 | 68.8 | 57.2 | 81.0 | 76.5 | 63.4 | 85.9 | 73.1 |
| PMATS* | **62.6** | **80.2** | **84.0** | 75.1 | **82.1** | 81.5 | **75.5** | **64.9** | **84.6** | **78.5** | **66.7** | **88.5** | **77.0** |

### 4.3 COMPARISON

Results on the Office-31 dataset are presented in Table 1. Our method with ResNet50 backbone obtains results similar to other best performing SoTA methods FixBi and RSDA-MSTN, around 91.4% average accuracy over all domain transfer tasks. Switching to the Deit-Small backbone, our accuracy improves by 1.0% on average, beating all other SoTA methods. Even CDTrans-S, which uses the same transformer backbone, only achieves a 90.4% accuracy compared to our 92.4%.

Table 2 shows our results on the Office-Home dataset. On this dataset, our method with the ResNet50 backbone performs slightly better than the SoTA CNN based method FixBi at 73.1% average accuracy. When we switch to a transformer backbone, however, we saw a significant improvement in accuracy across all categories. In fact, our PMATS* method has the best accuracy in most categories. The $P \rightarrow C$ and $R \rightarrow C$ categories show the largest improvement over the other SoTA methods, increasing by around 8% from the second best performing CDTrans, which also uses the Deit-Small backbone. Overall, our method beats the other SoTA methods by at least 2.3% With our large improvement in categories with lower baseline accuracies, this shows that our progressive mixup augmentation method achieves our goal of bridging large domain gaps to aid in the adaptation.

### 4.4 ABLATION STUDY

We conduct ablation studies on 4 of the 6 adaptation task of the Office-31 dataset to investigate the effectiveness of each of our components. In table 4.4, we show improvement through adding progressive mixup (PM), teacher-student (TS), and pseudo-label refinement (PR) over the baseline DSAN Zhu et al. (2020) with class aware sampling. Our progressive mixup improves the accuracy by an average of 1.5% while our teacher-student learning with two-stage pseudo-label refinement strategy provides an additional 0.9% improvement on average. This shows that each component is effective in improving performance.

Table 3: Ablation study on the effectiveness of our components on the four non-trivial tasks of the Office-31 dataset. We use Deit-Small as our backbone for this experiment.

| DSAN | PM | TS | PR | $A \rightarrow D$ | $A \rightarrow W$ | $D \rightarrow A$ | $W \rightarrow A$ | Avg |
|---|---|---|---|---|---|---|---|---|
| ✓ | | | | 94.4 | 96.0 | 77.7 | 78.6 | 86.7 |
| ✓ | ✓ | | | 96.0 | **97.7** | 78.9 | 80.1 | 88.2 |
| ✓ | ✓ | ✓ | | 96.6 | **97.7** | 79.5 | 80.6 | 88.6 |
| ✓ | ✓ | ✓ | ✓ | **97.4** | **97.7** | **80.4** | **80.7** | **89.1** |

## 5 CONCLUSION

In this paper, we propose a PMATS algorithm that aims to create intermediate augmented domains between the source and target domains, so as to aid in domain adaptation for cases with large domain discrepancy. We achieve this through progressive mixup between random source and target sample pairs to create intermediate samples that gradually shifts from source-like to target-like. By minimizing the discrepancy between our source and intermediate distributions, we are able to take advantage of multiple perspectives between the source and target domain during training. Furthermore, we propose a clustering-based pseudo-label refinement strategy combined with a temporal ensemble teacher network to generate more reliable target pseudo-labels for subdomain alignment. From our experimental results, we show our method to be superior to other SoTA UDA methods on difficult domain adaptation benchmarks.

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

## APPENDIX

## A  IMPLEMENTATION SETTINGS

For our backbone, we use ImageNet Deng et al. (2009) pretrained weights as initialization. Based on the class aware sampling procedure in Kang et al. (2019), in each iteration we randomly choose a subset of 10 classes to sample our our data from. For target data, we use our generated pseudo-label for the class aware sampling. Because our network initially cannot provide good target pseudo-labels, we define a warmup period of $W$ epochs during which we sample data normally.

In our experiments, we report our results using both ResNet50 He et al. (2016) and Deit-Small Touvron et al. (2021) as backbones for the Office-31 Saenko et al. (2010) and Office-Home Venkateswara et al. (2017) datasets to have a fair comparison other SoTA methods. More specifically, since the ResNet50 and Deit-Small have comparable model size, this yields a fair comparison with other methods that use either as a backbone. For both datasets, we set $\alpha$, $\beta$, and $d$ to be 0.5, 1, and 2 respectively for the mixup ratio beta distribution in Eq. (7). We use the same progressive schedule as in Zhu et al. (2020) for the trade-off term $\lambda_t$: $\lambda_{t\phi} = \frac{2}{(1+exp(-\gamma\theta))} - 1$ where $\theta$ is the training process linearly changing from 0 to 1 and $\gamma$ is 1. We use the SGD optimizer for training with a momentum of 0.9, an initial learning rate of 0.01, and a weight decay of 0.0005. Our learning rate schedule is defined as $n_\theta = \frac{n_0}{(1+a\theta)^b}$ where $n_0$ is the initial learning rate, $a$ is set to 2 for Office-31 and 3 for Office-Home, and $b$ is 0.75.

# B  CLUSTERING BASED PSEUDO-LABEL REFINEMENT

## B.1  K-MEANS CLUSTERING

After sending all of the target data through our model to obtain $z_j^t$ and $\hat{y}_j^t$, we compute initial cluster centroids $O_c$:

$$O_c = \frac{\sum_{j \in T} \hat{y}_j^t z_j^t}{\sum_{j \in T} \hat{y}_j^t}, \tag{10}$$

where $c$ is the category number and $T$ is the set of all target samples of size $N_t$. The one-hot pseudo-labels can then be reassigned via a nearest neighbor classifier, as shown in Eq. (11) using cosine distance in Eq. (12).

$$\hat{y}_{jc}^{*t} = \begin{cases} 1 & if\ c = argmin_k\ dist(O_k, z^t), \\ 0 & \text{otherwise,} \end{cases} \tag{11}$$

where

$$dist(a, b) = \frac{1}{2}(1 - \frac{<a, b>}{\|a\|\|b\|}). \tag{12}$$

Based on the new pseudo-labels $y^*$, we can calculate new cluster centroids:

$$O_c' = \frac{\sum_{j \in T} \hat{y}_j^{*t} z_j^t}{\sum_{j \in T} \hat{y}_j^{*t}}. \tag{13}$$

Equations (11) and (13) are repeated iteratively until convergence or the maximum number of clustering steps is reached.

## B.2  GAUSSIAN MIXTURE MODEL

Performing K-means gives us a hard pseudo-label since each target sample is directly assigned to one cluster with no probability of belonging to multiple clusters/classes. To convert this hard label to a soft probability distribution, we use a GMM with parameters initialized from our K-means clustering results from Section B.1. For the c-th Gaussian component, we can initialize the mean $\mu_c$, converiance $Cov_c$, and influence factor $\pi_c$ with the following equations.

$$\mu_c = \frac{\sum_{j \in T} \hat{y}_{jc}^{*t} z_j^t}{\sum_{j \in T} \hat{y}_{jc}^{*t}}, \tag{14}$$

$$Cov_c = \frac{\sum_{j \in T} \hat{y}_{jc}^{*t} (x_i^t - \mu_c)(x_i^t - \mu_c)^T}{\sum_{j \in T} \hat{y}_{jc}^{*t}}, \tag{15}$$

$$\pi_c = \frac{\sum_{j \in T} \hat{y}_{jc}^{*t}}{N_t}. \tag{16}$$

To fit the Gaussian model to our data, we use the expectation maximization (EM) algorithm Xiang & Wang (2003).

In the estimation (E) step, we use the currently estimated parameter values to estimate the probability that the data $x_j^t$ is generated by each component:

$$\gamma_{jc} = \frac{\pi_c N(x_j^t | \mu_c, Cov_c)}{\sum_{i=1}^{C} \pi_c N(x_j^t | \mu_c, Cov_c)}, \tag{17}$$

where $N(x | \mu, Cov)$ is the Gaussian density function.

In the maximization (M) step, we calculate the parameter values that would give the maximum likelihood expectation using Eqs. (14), (15), (16) to update $\hat{y}_{jc}^{*t}$ with $\gamma_{jc}$. The E and M steps are repeated until the value of the likelihood function converges or the maximum number of iterations is reached. With the finalized GMM parameters, we can use Eq (17) to also calculate the refined soft target pseudo-label $\hat{y}'^t$.

