# OpenReview forum: "Progressive Mixup Augmented Teacher-Student Learning for Unsupervised Domain Adaptation"
_ICLR.cc/2023/Conference — Submitted to ICLR 2023_

### Official Review · Reviewer_6xGt · 2022-10-17

**Confidence:** 4
**Correctness:** 2
**Technical Novelty And Significance:** 2
**Empirical Novelty And Significance:** 2
**Recommendation:** 3

**Clarity, Quality, Novelty And Reproducibility:**

Quality:

The presentation of this paper needs improvement. For example, the resolution of Figure 1 is not satisfactory, and the font used in figures looks not professional.


Novelty:

The proposed method looks like a simple combination of existing techniques. Authors should better manifest the novel technical contribution of their method.


Reproducibility:

Authors did not submit the source codes, and thus the reproducibility is not guaranteed.


**Strength And Weaknesses:**

Strength:

1. In general, the proposed techniques (i.e., scheduled domain mixup and pseudo labeling with teacher network) are valid to perform progressive domain alignment between source and target domains.


Weakness:

1. Most of the proposed techniques have been studied in previous works, including domain mixup [a], momentum encoder for pseudo labeling [b] and clustering-based classification refinement [c]. Therefore, authors should better present which parts of the proposed method are novel and which parts follow previous works. Authors are suggested to add a subsection to the current draft to discuss the connection and difference against these technically related works.
2. The experimental verification is not sufficient. It is glad to see improvements on Office-31 and Office-Home. However, considering the development of UDA at scale, the audience would definitely like to see the results on larger-scale UDA datasets under diverse domain shifts. It is unknown how the proposed method performs on the datasets like DomainNet and VisDA?



[a] Xu, Minghao, et al. "Adversarial domain adaptation with domain mixup." Proceedings of the AAAI Conference on Artificial Intelligence. Vol. 34. No. 04. 2020.

[b] Huang, Jiaxing, et al. "Category contrast for unsupervised domain adaptation in visual tasks." Proceedings of the IEEE/CVF Conference on Computer Vision and Pattern Recognition. 2022.

[c] Tang, Hui, Ke Chen, and Kui Jia. "Unsupervised domain adaptation via structurally regularized deep clustering." Proceedings of the IEEE/CVF conference on computer vision and pattern recognition. 2020.


**Summary Of The Paper:**

This paper proposes the progressive mixup augmented teacher-student (PMATS) training strategy to tackle unsupervised domain adaptation (UDA). In specific, a momentum encoder based teacher network is employed to assign pseudo labels to target domain samples, and a student network is trained to classify source domain samples and align the distribution of the source and the intermediate mixup domains. A mixup scheduling scheme is designed to progressively move the mixup domain from source to target along training. Authors demonstrate the performance gains of PMATS on Office-31 and Office-Home datasets.

**Summary Of The Review:**

In summary, the current manuscript does not meet the high standard of ICLR on technical novelty, experimental verification and presentation. I give a rejection currently and recommend improving the paper according to the weakness.

---

### Official Review · Reviewer_dVDV · 2022-10-26

**Confidence:** 5
**Correctness:** 3
**Technical Novelty And Significance:** 1
**Empirical Novelty And Significance:** 1
**Recommendation:** 3

**Clarity, Quality, Novelty And Reproducibility:**

This paper is clear. The presentation quality is moderate. The novelty is limited.

**Strength And Weaknesses:**

### Strength
1. The proposed method is technically sound.
2. Experiments on two datasets verify the effectiveness

### Weakness
1. The novelty is limited. Mean teacher models have been used by quite a few works to address the UDA problem, e.g., [1,2,3,4]. I do not see a clear novelty in the proposed method of using a mean model to address UDA. Mixup is also a common technique to enhance generalizablity, and has been used to address UDA before [5,6]. I do not see a strong novelty here as well.
2. The evaluation is not sufficient and convincing. Only two datasets are employed, while established practices in this area use at least three datasets for evaluation.


- [1] MTTrans: Cross-Domain Object Detection with Mean Teacher Transformer
- [2] Unbiased Mean Teacher for Cross-domain Object Detection
- [3] Exploring Object Relation in Mean Teacher for Cross-Domain Detection
- [4] CMT: Cross Mean Teacher Unsupervised Domain Adaptation for VHR Image Semantic Segmentation
- [5] Adversarial Domain Adaptation with Domain Mixup
- [6] Dual Mixup Regularized Learning for Adversarial Domain Adaptation

**Summary Of The Paper:**

This paper address UDA with the mean teacher model where the teacher model generate pseudo labels to supervise the student model, and the teacher model is updated as the moving average of the student model. Mixup is adopted to help enhance generalization performance. Experiments on two datasets verify the effectiveness.

**Summary Of The Review:**

This paper is below bar of this conference for the limited technical contribution and unconvincing experimental evaluation.

---

### Official Review · Reviewer_CyYH · 2022-10-26

**Confidence:** 3
**Clarity, Quality, Novelty And Reproducibility:** The novelty of this paper may be insu…
**Correctness:** 2
**Technical Novelty And Significance:** 1
**Empirical Novelty And Significance:** 1
**Recommendation:** 3

**Strength And Weaknesses:**

Strength:
- The empirical performance is strong.

Weaknesses:
- This paper actually proposes a combination of some existing engineering techniques (e.g., EMA [*3], mixup [*1, *2], pseudo-label refinement [*4], Kmeans, and GMM), and tunes a group of hyper-parameters, yielding a strong performance on common benchmarks. The novelty and scientific contributions are trivial. This paper does not provide sufficient insights on the design of the proposed method. Personally, it is more like an experimental report instead of a paper.
- At least several questions need to be answered. Why do we need the teacher-student architecture? Why do we need to refine the pseudo label? Why do we need to perform mixup? The simple "it improves the accuracy" is not a proper answer to these questions.
- After reading the reviews from other reviewers and carefully reading the related works [*1-*4], I find that the aforementioned questions have been largely answered in [*1-*4]. Hence, now my major concern lies in the novelty of this paper. I think at least the authors should discuss the difference between their work and [*1-*4]. If there are some differences, the authors may consider investigating replacing their design with the techniques in [*1-*4], reporting the empirical changes, and analyzing why this happens.

[*1] Adversarial domain adaptation with domain mixup. AAAI 2020.

[*2] Dual Mixup Regularized Learning for Adversarial Domain Adaptation. ECCV 2020.

[*3] Category contrast for unsupervised domain adaptation in visual tasks. CVPR 2022.

[*4] Unsupervised domain adaptation via structurally regularized deep clustering. CVPR 2020.

**Summary Of The Paper:**

This paper proposes a strong engineering solution for the task of Unsupervised Domain Adaptation (UDA). The proposed method includes a student model and a teacher model. The latter is obtained by performing an exponential moving average on the former. The pseudo labels of the images in the target domain are obtained by refining the prediction of the teacher model. The authors further leverage the mixup augmentation to improve the performance.

**Summary Of The Review:**

I think the current paper is not ready to be published on ICLR. See the weaknesses above.

---

### Official Review · Reviewer_kMVd · 2022-10-31

**Confidence:** 4
**Correctness:** 3
**Technical Novelty And Significance:** 2
**Empirical Novelty And Significance:** 2
**Recommendation:** 5

**Clarity, Quality, Novelty And Reproducibility:**

Clarity: The paper is written in a clear manner. As I was reading the paper, I was able to understand the main ideas clearly.

Quality and novelty: I feel that the novelty and quality is a bit lacking. The method merely combines ideas proposed in several other paper, but does not do a good job of justifying the design choice. Experimental validation is also not strong.

Reproducibility: Some implementation details are provided in appendix. But given the complex nature of this system, I am a bit concerned if it would be easy to reproduce the results.

**Strength And Weaknesses:**

The approach seems to combine several ideas from UDA which seems to work well and get the SOTA on some benchmark datasets.

This paper is what seems like several ideas from UDA put together in one framework. All these ideas - pseudo labeling, mixup augmentation, teacher-student approaches and LMMD have all been proposed separately and used in other papers. This paper merely combines all these approaches. While there is some merit in doing that, I feel like the paper has not done a good job of comparing how and why their combined approach improves over each of the independent methods. There is an ablation showing that the performance improves, but other than that, there is no proper visualization / justification as to why this is better. I can think of other ways of combining the approaches, but why this specific way? What happens to the feature space due to each of the loss terms? Why does each loss term help? Some visualizations explaining these might help the readers understand the method a bit better.

I feel like the paper has not compared with some of the best SOTA approaches on these benchmarks. For instance, [1] achieves a performance of 73% and [2] achieves 72.2% accuracy on Office-Home benchmark with Resnet-50 backbone. These papers are not compared. Compares to these numbers, this paper only achieves a 0.1% performance improvement. The paper achieves 77% with DeiT backbone, claiming that as SOTA is not a but fair because with improved backbone, these approach might also get good performance.

One of my main concerns with this paper is the lack of evaluation on some hard domain adaptation benchmarks such as DomainNet. The performance on Office-31 is already saturated and the current methods only obtain marginal gains. DomainNet, on the other hand, is a comprehensive benchmark for evaluating adaptation algorithms.

The paper has so many components which make the system a bit complicated. I wonder whether this system would help in practice, or if there would be many hyperparameters to tune.

[1] Liu et al., "Cycle Self-Training for Domain Adaptation", NeurIPS 2021
[2] Prabhu et al., "Selective entropy optimization via committee consistency for unsupervised domain adaptation", ICCV 2021.


**Summary Of The Paper:**

This paper proposes a method for unsupervised domain adaptation wherein pseudo-labeling, teacher-student approaches, augmentation (mix-up) and MMD based alignment are put together in one framework. Given a target sample and source sample, a mix-up sample is first constructed. The label for the mix-up sample is created using a pseudo-label obtained from the teacher network. With the source, mixup data-target pair, an alignment loss based on LMMD and a source classification loss is used for updating the student network. The teacher network is a EMA of student network.  This approach seems to achieve good performance on some benchmark datasets.

**Summary Of The Review:**

I feel like this is one of the papers that merely combines several ideas already used in domain adaptation, but does not do a good job of thoroughly studying their approach. Experimental validation is also weak. So, I am inclined towards rejecting the paper.

---

### Official Review · Reviewer_XSGj · 2022-11-05

**Confidence:** 4
**Correctness:** 2
**Technical Novelty And Significance:** 2
**Empirical Novelty And Significance:** 2
**Recommendation:** 3

**Clarity, Quality, Novelty And Reproducibility:**

The explanation for the detailed framework is not clear, and the algorithm seems incremental from the previous studies. The problem setting is equivalent to the conventional UDA task, and its solution is relatively small from the other work. Due to the lack of a detailed explanation, we cannot reproduce the algorithm.

**Strength And Weaknesses:**

The abstract and the experimental validation is written well, so the motivation of the proposed algorithm and its validation were successfully delivered. However, the detailed explanation of the methodology contains many typos and errors, which reduces the reliability of the designed framework. In addition, I have several questions and feedback as follows:

1. Lack of experiments for large-scale dataset
 The proposed algorithm was validated by using two UDA benchmarks including Office-31 and Office-Home. Even though we can validate the domain-variant robustness of the UDA algorithms by using the two datasets, the datasets have an insufficient number of training samples to show the scale robustness. I recommend the authors perform additional experiments to show its scale robustness by using large-scale datasets such as the VisDA dataset or UDA segmentation scenarios. If the algorithm is validated through the segmentation task, we can verify the real-world applicability of the proposed algorithm also.

2. Qualitative validation of gradual change
  The authors insist that the proposed algorithm can generate the intermediate images and the intermediate images change gradually during the training phase. Unfortunately, it is hard to imagine the gradual change of intermediate samples just through the quantitative results. Although several graphical examples are visualized in some frameworks, qualitative examples would be more helpful to understand the meaning of the gradual change of the intermediate sample.

3. Mechanism to increase the model robustness to noisy pseudo-labels
  As I understand, the model becomes robust to the noisy pseudo-labels by using LMMD, but the LMMD is referred to in the previous study. Without the LMMD, only the intermediate sample using mixup augmentation is a novel approach to this framework. Then, which new module is helpful to increase the robustness of the noisy pseudo-labels? Or, the synergy between the LMMD and PMATS is special? If then, why?

4. Difference with "UDA with progressive domain augmentation", 2020
 The overall approach and framework are very similar to those of "UDA with progressive domain augmentation". The only difference with the previous work is to utilize the mixup augmentation. I hope to recognize the novelty of the previous work.

5. Role of p and q' in Eq. 6
  Even though the function d_H is fed by p and q', there is no related notation in the following equation. We need more explanation for the equation.

6. Errors in Eq 2 and Eq 5
  In Eq.2, there is no "=", and the right part of the norm parenthesis is missing in Eq. 5. The first character after Eq.5 should be lower-cased (w).

7. Motivation and analysis for d, alpha, and beta
  Even though the authors present the previous approaches to determine the hyperparameters of mixup, the motivation of the proposed sampling mechanism is not exactly explained. Why is the new sample mechanism necessary? With this motivation, the empirical analysis for its validity should be added also.

**Summary Of The Paper:**

This paper presents a framework to reduce the domain gap between the source and target domains in the unsupervised domain adaptation task. The teacher network is used to increase the reliability of pseudo-labels, and the progressive mixup augmentation strategy is employed to generate the intermediate samples to bridge the large domain gap. This training strategy is called PMATS, which shows the state-of-the-art performance of the two UDA benchmark datasets.

**Summary Of The Review:**

Even with the state-of-the-art performance, the insufficient description and the incremental framework are improper for the publication. The authors should improve the description of the motivation and the detailed framework for the next submission.

---

### Decision · Program_Chairs · 2023-01-20

**Decision:**

Reject

**Justification For Why Not Higher Score:**

All reviewers recommended reject. The paper has major issues with technical novelty and empirical evulations.

**Justification For Why Not Lower Score:**

N/A

**Metareview: Summary, Strengths And Weaknesses:**

The paper describes mixup training for improving a teacher-student approach for UDA.

Reviewers found that the contribution is limited since the paper integrates several existing ideas. In terms of empirical contribution, reviewers pointed to missing experiments with large and challenging benchmarks, and to missing baselines. All four reviewers recommended rejection.

No rebuttal was submitted.

**Summary Of Ac-Reviewer Meeting:**

N/A